



# Three-dimensional normal mode functions: Open access tools for their computation in isobaric coordinates (p-3DNMF.v1)

Carlos A. F. Marques[1], Martinho Marta-Almeida[2,3], and José M. Castanheira[1]

[1]CESAM & Departamento de Física, Universidade de Aveiro. Aveiro, Portugal
[2] Universidade de Vigo, Campus Lagoas-Marcosende, Vigo, Spain
[3]Centro Oceanográfico de A Coruña, Instituto Español de Oceanografía, A Coruña, Spain

**Correspondence:** J. M. Castanheira (jcast@ua.pt)

**Abstract.**

A free software package for the computation of the 3-Dimensional Normal Modes of an hydrostatic atmosphere is presented. This software performs the computations in isobaric coordinates and was developed for two user friendly languages: MATLAB and Python. The software can be used to expand the global atmospheric circulation onto the 3-D Normal Modes. This expansion allows the computation of a 3-D energetic scheme which partition the energy reservoirs and energy interactions between 3-D spatial scales, and between barotropic and baroclinic components as well as between balanced (rotational) and unbalanced (divergent) circulation fields. Moreover, by retaining only a subset of the expansion coefficients, the 3-D normal mode expansion can be used for spatial scale filtering of atmospheric motion, and filtering of balanced motion and mass unbalanced motions, and barotropic and baroclinic components. Fixing the meridional scale, the 3-D normal mode filtering can be used to isolate tropical components of the atmospheric circulation. All these features are useful both in data analysis and in assessment of general circulation atmospheric models.

## 1 Introduction

The use of the three-dimensional normal mode functions (3-D NMFs) of the linearized primitive equations as a basis to expand the global horizontal wind and mass fields simultaneously was presented for the first time by Kasahara and Puri (1981). The expansion of both atmospheric fields onto the 3-D NMFs allows the partition of energy of the global motion in two kinds of motions: mass balanced rotational modes (Rossby/Hawitz waves) and divergent modes (inertio-gravity waves).

The original work of Kasahara and Puri (1981) was developed for terrain following sigma-coordinates. Four years later, Tanaka (1985) extended the methodology to isobaric coordinates. In these coordinates, Tanaka (1985) and Tanaka and Kung (1988) were able to develop a 3-D normal mode energy scheme, including the exchanges of Kinetic and Available Potential Energy (APE) among the different 3-dimensional scales and types of motions. Recently, Marques and Castanheira (2012) extended the methodology to analyse the conversion of APE into Kinetic energy.

The 3-D normal mode expansion has been applied in several type of studies, such as the comparisons of reanalysis datasets (Marques and Castanheira, 2012; Yamagami and Tanaka, 2016, and references therein), or the analysis of ensemble prediction system uncertainties (Žagar et al., 2015a). The methodology was also applied in studies of low frequency extratropi-





cal variability, like the Arctic/North Atlantic Oscillation (AO/NAO) (Castanheira et al., 2002; Tanaka and Tokinaga, 2002; Tanaka and Seki, 2013; Castanheira and Marques, 2019) as well as in tropical variability analysis (Yamagami and Tanaka, 2014; Castanheira and Marques, 2015; Marques and Castanheira, 2018; Blaauw and Žagar, 2018; Franzke et al., 2019; Kitsios et al., 2019).

5     Recently, Žagar et al. (2015b) developed an open source software package for the projection of global horizontal wind and mass fields onto 3-D NMFs. Their software was developed for sigma coordinates, but does not allow for the analysis of the full energy cycle. Here we present a software package for the projections of horizontal wind and mass fields onto 3-D NMFs, using isobaric coordinates. This software allows for the analysis of the full atmospheric energy cycle and was developed for two user friendly languages: MATLAB and Python. The methodology described in this study to obtain the 3-D NMFs

10  has been used in previous works (e.g. Marques and Castanheira, 2012; Marques et al., 2014; Castanheira and Marques, 2015; Marques and Castanheira, 2018; Castanheira and Marques, 2019) in the form of unstructured and disperse scripts written for the specific data set used in those studies. The software presented here consists of a structured collection of functions, intended to be used for a wider and more general data sets (such as reanalysis and outputs from climate models) and requiring the minimum user interaction as we found possible.

## 2   3–D normal-mode functions

### 2.1   The basic equations

In the traditional shallow-atmosphere approximation, a set primitive equations in isobaric coordinates may be written as

$$\frac{\partial u}{\partial t} - 2\Omega v \sin\theta + \frac{1}{a\cos\theta}\frac{\partial \phi}{\partial \lambda} = -\boldsymbol{V}\cdot\nabla u - \omega\frac{\partial u}{\partial p} + \frac{\tan\theta}{a}uv + F_u, \tag{1}$$

$$\frac{\partial v}{\partial t} + 2\Omega u \sin\theta + \frac{1}{a}\frac{\partial \phi}{\partial \theta} = -\boldsymbol{V}\cdot\nabla v - \omega\frac{\partial v}{\partial p} - \frac{\tan\theta}{a}u^2 + F_v, \tag{2}$$

$$\frac{\partial \phi}{\partial p} = -\frac{RT}{p}, \tag{3}$$

$$\nabla \cdot \boldsymbol{V} + \frac{\partial \omega}{\partial p} = 0, \tag{4}$$

$$\frac{\partial T}{\partial t} - \frac{pS_0}{R}\omega = -\boldsymbol{V}\cdot\nabla T - \omega\frac{\partial T}{\partial p} + \frac{\dot{Q}}{c_p}. \tag{5}$$

where $\lambda$ and $\theta$ are longitude and latitude, respectively. Variables $u$ and $v$ are the zonal and meridional components of horizontal wind vector $\boldsymbol{V}$; and the vertical wind component in the isobaric system is represented by $\omega$. The constants $\Omega$ and $a$ denote the Earth's angular velocity and Earth's radius, respectively. Variable $\dot{Q}$ is the rate of diabatic heating per mass unity, $c_p$ the specific

30  heat at constant pressure, $p$, and $R$ is the dry air gas constant. Temperature and geopotential, $T$ and $\phi$, are the deviations from



a hydrostatic reference state $T_0(p)$ and $\phi_0(p)$, respectively; and the static stability parameter, $S_0$, is given by

$$S_0 = \frac{R}{p}\left(\frac{RT_0}{pc_p} - \frac{\mathrm{d}T_0}{\mathrm{d}p}\right). \tag{6}$$

Multiplying (5) by $R/(pS_0)$, then calculating the derivatives with respect to $p$, and finally using on the resulting equation the hydrostatic and continuity equations ((3) and (4)), the thermodynamic energy equation takes the form

$$-\frac{\partial}{\partial t}\frac{\partial}{\partial p}\left(\frac{1}{S_0}\frac{\partial \phi}{\partial p}\right) + \nabla \cdot \boldsymbol{V} = \frac{\partial}{\partial p}\left[\frac{R}{pS_0}\left(-\boldsymbol{V}\cdot\nabla T - \omega\frac{\partial T}{\partial p}\right)\right] + \frac{\partial}{\partial p}\left(\frac{R\dot{Q}}{pc_pS_0}\right). \tag{7}$$

The right-hand side of equations (1), (2) and (7) contain the nonlinear terms, frictional forces and the diabatic heat sources. By setting the right-hand sides of those equations to zero, the linearised system of equations for the three dependent variables $u$, $v$ and $\phi$ is written as

$$\frac{\partial u}{\partial t} - 2\Omega v\sin\theta + \frac{1}{a\cos\theta}\frac{\partial \phi}{\partial \lambda} = 0, \tag{8}$$

$$\frac{\partial v}{\partial t} + 2\Omega u\sin\theta + \frac{1}{a}\frac{\partial \phi}{\partial \theta} = 0, \tag{9}$$

$$-\frac{\partial}{\partial p}\left[\frac{1}{S_0}\frac{\partial}{\partial p}\left(\frac{\partial \phi}{\partial t}\right)\right] + \nabla \cdot \boldsymbol{V} = 0. \tag{10}$$

The linearised system (8)-(10) describes small oscillations of an incompressible, homogeneous, hydrostatic and inviscid fluid over a rotating sphere, around a state at rest (Tanaka and Kung, 1988; Daley, 1991). Assuming the following separation of variables (Kasahara and Puri, 1981; Daley, 1991),

$$[u, v, \phi] = \Psi(p)\left[\tilde{u}(\lambda,\theta,t), \tilde{v}(\lambda,\theta,t), \tilde{\phi}(\lambda,\theta,t)\right], \tag{11}$$

and substituting into (10), one obtains

$$\frac{1}{\Psi}\frac{\partial}{\partial p}\left(\frac{1}{S_0}\frac{\partial \Psi}{\partial p}\right) = \frac{\nabla \cdot \tilde{\boldsymbol{V}}}{\partial\tilde{\phi}/\partial t} = -\frac{1}{gh}. \tag{12}$$

where $g$ is the constant gravity acceleration and $1/(gh)$ is the separation constant.

Then the separable vertical structures $\Psi$ of the solutions of the linearised system (8)-(10) are given by the eigensolution of the following vertical structure equation (VSE)

$$\frac{\partial}{\partial p}\left(\frac{1}{S_0}\frac{\partial \Psi}{\partial p}\right) + \frac{1}{gh}\Psi = 0. \tag{13}$$

satisfying an appropriate pair of upper and lower boundary conditions.

## 2.2 Vertical structure functions

The VSE (13) with appropriate boundary conditions form an eigenvalue problem, with discrete eigensolutions $\Psi_k(p)$ and eigenvalues $-1/gh_k$ (Cohn and Dee, 1989). The software here presented was developed for two sets of boundary conditions.





One set was derived from the linearised version of the thermodynamic equation with the assumption that the vertical $p$-velocity, $\omega$, vanishes at the top of the atmosphere, and from the assumption that the linearised geometric vertical velocity $w = \mathrm{d}z/\mathrm{d}t$ vanishes at a constant pressure, $p_s$, near the surface (Tanaka, 1985). Such boundary conditions may be written as

$$\frac{\mathrm{d}\Psi_k}{\mathrm{d}p} + \frac{pS_0}{RT_0}\Psi_k = 0 \qquad \text{at} \qquad p = p_s \tag{14}$$

$$\lim_{p \to 0} \frac{1}{S_0}\frac{\mathrm{d}\Psi_k}{\mathrm{d}p} = 0. \tag{15}$$

Another set of boundary conditions assumes that the atmospheric mass is bounded by an isobaric surface $p_s$, and the pressure vertical velocity $\omega$ must also vanish at $p_s$. In this case, the lower boundary condition (14) becomes

$$\frac{\mathrm{d}\Psi_k}{\mathrm{d}p} = 0 \quad \text{at} \quad p = p_s. \tag{16}$$

10    The vertical structure functions (VSFs), $\Psi_k(p)$, can be normalized to satisfy the following orthonormality condition

$$\frac{1}{p_s}\int_0^{p_s} \Psi_i(p)\Psi_j(p)\mathrm{d}p = \delta_{ij}. \tag{17}$$

Using this orthonormality condition, one can define a vertical transform

$$f_k = \frac{1}{p_s}\int_0^{p_s} f(p)\Psi_k(p)\mathrm{d}p, \tag{18}$$

where $f(p)$ is an arbitrary function of pressure.

15    It can be shown (Cohn and Dee, 1989) that the VSFs form a complete orthonormal basis and the circulations variables $(u, v, \phi)^T$, may be expanded as

$$(u, v, \phi)^{\mathrm{T}} = \sum_{k=0}^{\infty} (u_k, v_k, \phi_k)^{\mathrm{T}} \Psi_k(p), \tag{19}$$

where

$$(u_k, v_k, \phi_k)^{\mathrm{T}} = \frac{1}{p_s}\int_0^{p_s} (u, v, \phi)^{\mathrm{T}} \Psi_k(p) \, \mathrm{d}p, \tag{20}$$

20    In this software package, the VSE is solved by using the spectral method introduced by Kasahara (1984) (see also Castanheira et al. (1999)), which has the advantage, over the finite difference method, that the derivatives of the vertical structure function can be calculated by analytical differentiation. The input data is the temperature profile $T_0$, which is assumed to be defined on $N_k$ pressure levels. Following the computational results of Castanheira et al. (1999), the number $J$ of Legendre polynomials used to approximate $\Psi$ is defined as $J = N_k + 20$. The vertical integration is performed by Gaussian quadrature interpolating $T_0$





into $G_L = 2J - 1$ Gaussian levels. The maximum number of VSFs saved in the output is equal to the number of pressure levels ($N_k$).

Fig. 1 shows the vertical profiles of temperature $T_0$ and base-10 logarithm of the stability parameter $S_0$. The profiles were computed based on the 3–D temperature field of the ERA-interim reanalysis (Dee et al., 2011) covering the period from 1

5 January, 1979 to 31 December, 2010. The 3–D temperature field was obtained with $1.5°$lon. $\times 1.5°$lat. grid resolution for the 37 isobaric levels from 1 to 1000 hPa, at time intervals of six-hours. The first 12 vertical structures associated with the 12 smallest eigenvalues $\xi_k$, which were obtained using boundary conditions (14) and (15), are shown in Fig. 2. The first 12 vertical structures obtained using boundary conditions (16) and (15), are illustrated in Fig. 3. Both sets of 37 VSFs were derived using the temperature profile $T_0$ and stability parameter $S_0$ represented in Fig. 1.

10 The baroclinic structures in both figures 2 and 3, i.e. the vertical structures with one or several nodes are close to each other. In Fig. 3 the vertical structure function, $\Psi_0$, is a pure barotropic vertical structure, i.e. it is a constant, and the projection of an atmospheric variable onto the barotropic mode correspond exactly to the mass weighted vertical mean of that variable. The vertical structure function $\Psi_0$ in Fig. 2 is not strictly constant. Nevertheless, since $\Psi_0$ of Fig. 2 does not change sign as its counterpart in Fig. 3, and is approximately constant in the troposphere, it is also called as barotropic mode.

## 2.3 Computation of the horizontal structure functions

Applying the vertical transform (18) to the system of equations (8)-(10), a dimensionless equation is obtained in the following vector form (Tanaka, 1985; Marques and Castanheira, 2012):

$$\frac{\partial}{\partial \hat{t}} \mathbf{W}_k + \mathbf{L} \mathbf{W}_k = 0, \tag{21}$$

where

$$\hat{t} = 2\Omega t, \tag{22}$$

$$\mathbf{W}_k = \mathbf{X}_k^{-1} \begin{bmatrix} u \\ v \\ \phi \end{bmatrix}_k, \tag{23}$$

$$\mathbf{L} = \begin{bmatrix} 0 & -\sin\theta & \frac{\alpha_k}{\cos\theta}\frac{\partial}{\partial\lambda} \\ \sin\theta & 0 & \alpha_k\frac{\partial}{\partial\theta} \\ \frac{\alpha_k}{\cos\theta}\frac{\partial}{\partial\lambda} & \frac{\alpha_k}{\cos\theta}\frac{\partial}{\partial\theta}(\cos\theta\,(\cdot)) & 0 \end{bmatrix}. \tag{24}$$



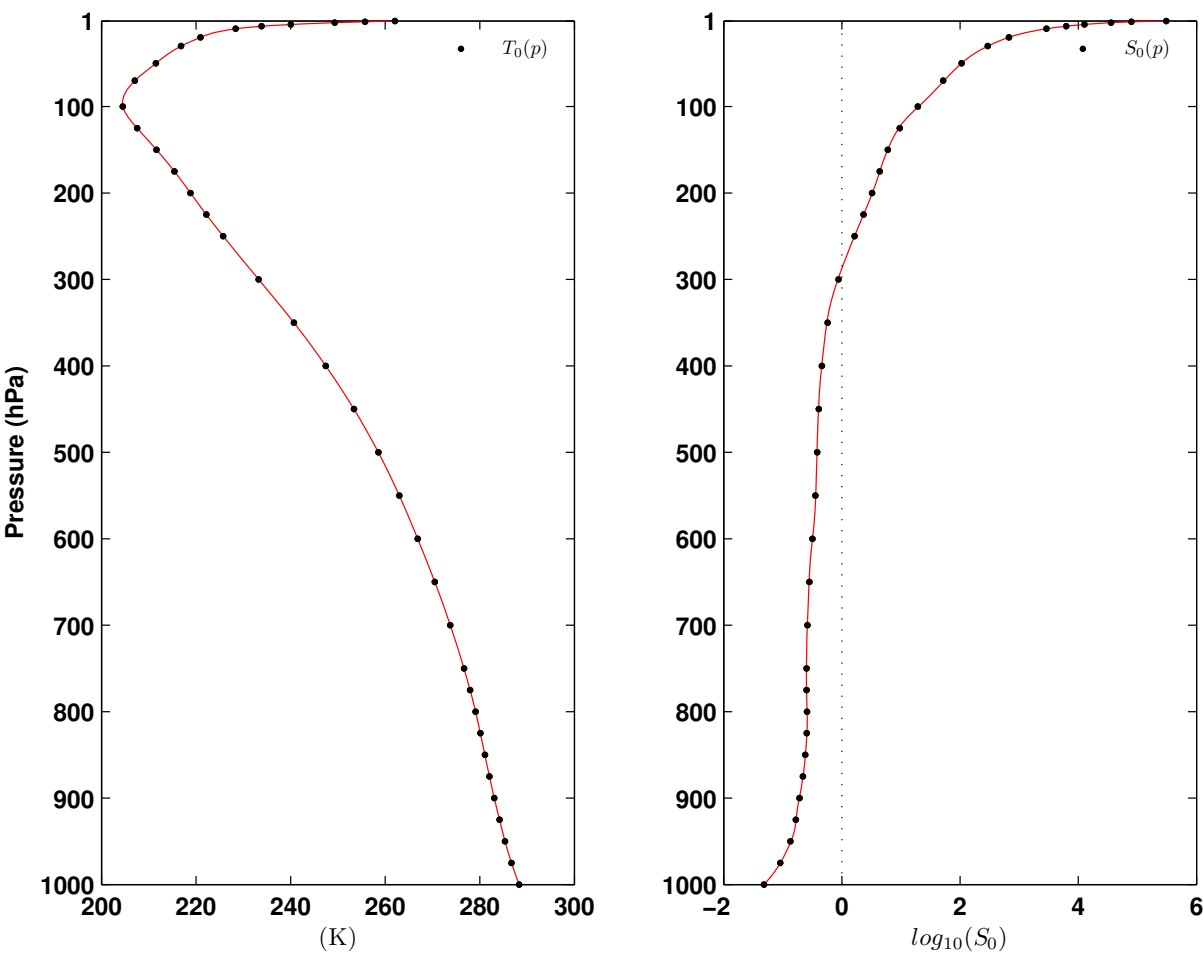

**Figure 1.** Left: Vertical profile of temperature at the ERA-interim pressure levels (black dots) and interpolated into $G_L$ gaussian levels (red line). Right: stability parameter $S_0$ calculated from $T_0$ at original pressure levels (black dots) and from interpolated $T_0$ at gaussian levels (red line). The stability parameter $S_0$ was made dimensionless multiplying eq. (6) by $p_s^2/gH_*$ with $H_* = 8$ km. The profiles were computed from the 3–D temperature field of the ERA-interim reanalysis covering the period from 1 January, 1979 to 31 December, 2010.





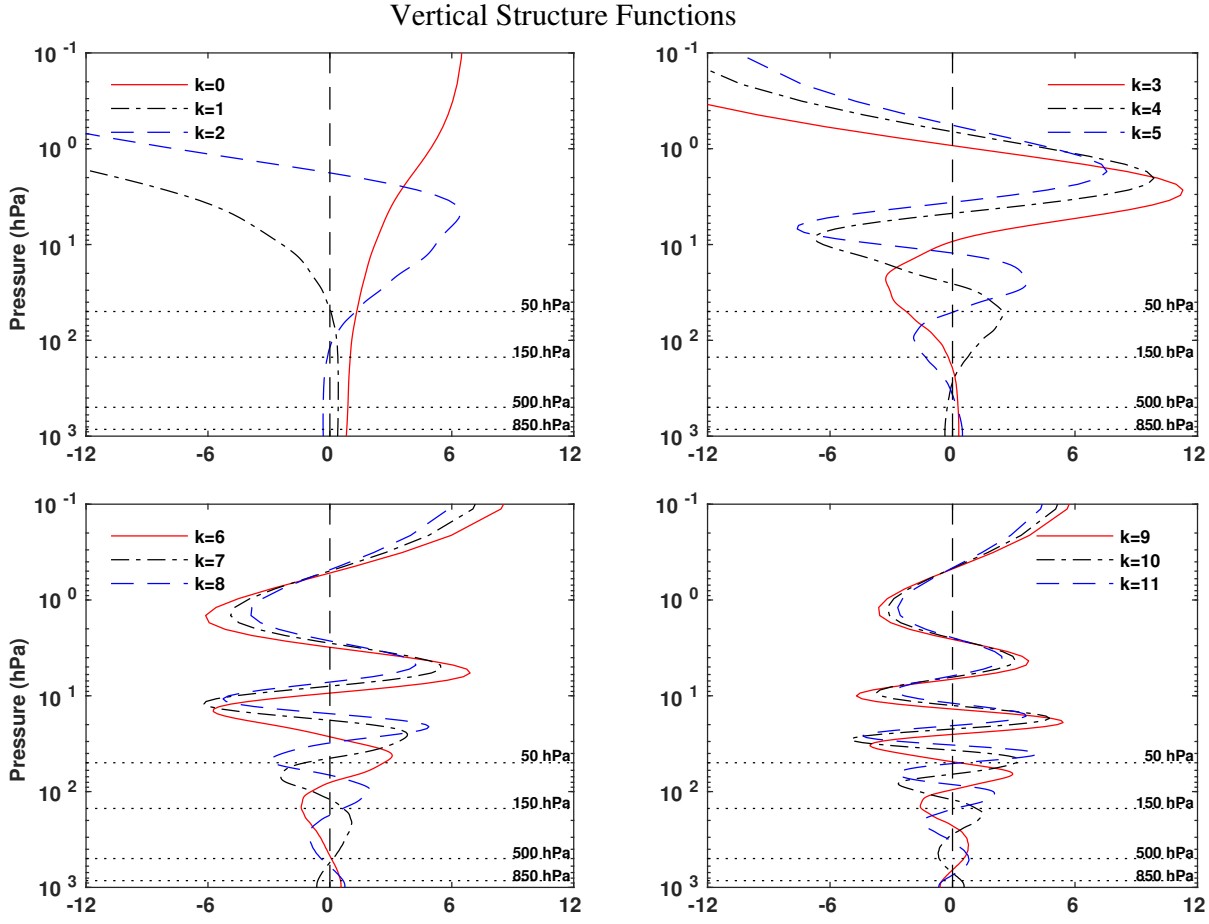

**Figure 2.** First 12 vertical structure functions derived using $T_0$ and $S_0$ of Fig. 1, with boundary conditions (14) and (15).

The time was made dimensionless by multiplying with $2\Omega$, and the vectors are made dimensionless by multiplying by the inverse of the scaling matrix $\mathbf{X}_k$, which is given by

$$\mathbf{X}_k = \begin{bmatrix} \sqrt{gh_k} & 0 & 0 \\ 0 & \sqrt{gh_k} & 0 \\ 0 & 0 & gh_k \end{bmatrix}. \tag{25}$$

The dimensionless parameter, $\alpha_k$, in the linear differential matrix operator $\mathbf{L}$ is defined as

$$\alpha_k = \frac{\sqrt{g\,h_k}}{2\,\Omega\,a}. \tag{26}$$

Equation (21) is a dimensionless form of the linearized shallow-water equations.



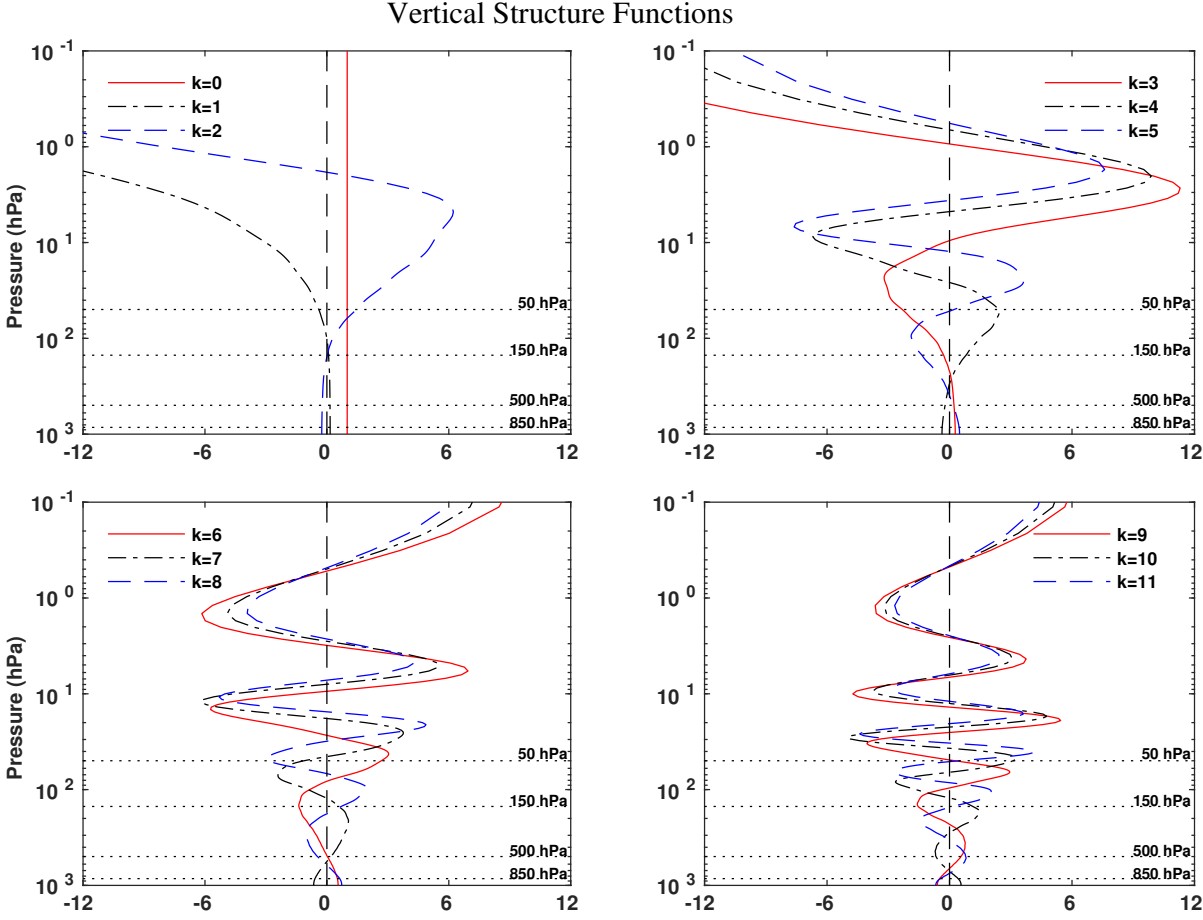

**Figure 3.** First 12 vertical structure functions derived using $T_0$ and $S_0$ of Fig. 1, with boundary conditions (16) and (15).

Since equation (21) is a linear system with respect to $\lambda$ and $\hat{t}$, the solution $\mathbf{W}_k$ can be expressed as zonal waves (Swarztrauber and Kasaha 1985)

$$\mathbf{W}_{nlk}\left(\lambda,\theta,\hat{t}\right) = \mathbf{\Theta}_{nlk}\left(\theta\right) e^{i\left(n\lambda - \sigma_{nlk}\hat{t}\right)}, \tag{27}$$

where $\sigma_{nlk}$ are the dimensionless frequencies for the free waves with zonal wave number $n$ and meridional structures $\mathbf{\Theta}_{nlk}\left(\theta\right)$.

5   The functions $\mathbf{H}_{nlk}(\lambda,\theta) = \mathbf{\Theta}_{nlk}\left(\theta\right) e^{i\,n\,\lambda}$ are known as the *horizontal structure functions*, and the meridional structures $\mathbf{\Theta}_{nlk}$ are the Hough vectors functions (Longuet-Higgins, 1968; Swarztrauber and Kasahara, 1985). Substituting (27) into (21) gives

$$-i\,\sigma_{nlk}\mathbf{H}_{nlk}\left(\lambda,\theta\right) + \mathbf{L}\,\mathbf{H}_{nlk}\left(\lambda,\theta\right) = 0, \tag{28}$$

and thus the *horizontal structure functions* are obtained as free eigenvalue problem. Equation (28) is referred as the horizontal structure equation (HSE).



For $n > 0$, there are two kinds of motion (i.e. solutions) with distinct frequencies for system (21). The first kind describes high-frequency eastward and westward propagating gravity-inertia waves, and the second kind describes low-frequency westward propagating rotational waves of the Rossby-Haurwitz type (Longuet-Higgins, 1968; Kasahara and Puri, 1981; Swarztrauber and Kasahara, 1985; Žagar et al., 2015b).

It can be shown that for real $\alpha_k$ all the frequencies $\sigma$ of $\mathbf{L}$ are real and that any modes $\mathbf{H}$ corresponding to distinct frequencies are orthogonal (e.g. Swarztrauber and Kasahara, 1985; Žagar et al., 2015b). For $n > 0$, the frequencies are distinct and thus the modes are orthogonal. However, for zonal wavenumber $n = 0$, the rotational modes are not necessarily orthogonal because their frequencies are all zero. Nevertheless, it is possible to derive an orthogonal set of rotational modes for $n = 0$, which are also orthogonal to the modes for $n > 0$ (Kasahara and Puri, 1981; Shigehisa, 1983; Swarztrauber and Kasahara, 1985).

Therefore, all horizontal structure functions $\mathbf{H}$ satisfy the orthonormal condition given by

$$\frac{1}{2\pi} \int\limits_{-\frac{\pi}{2}}^{\frac{\pi}{2}} \int\limits_{0}^{2\pi} \mathbf{H}^*_{nlk} \cdot \mathbf{H}_{n'l'k} \cos\theta \mathrm{d}\lambda \mathrm{d}\theta = \delta_{nn'} \delta_{ll'}, \tag{29}$$

where the superscript $*$ denotes a conjugate transpose and the right-hand side is unity if $n = n'$ and $l = l'$, and zero otherwise.

The Hough vector function, $\mathbf{\Theta}_{nlk}(\theta)$, has three components, zonal velocity, meridional velocity and height, and is given by

$$\mathbf{\Theta}_{nlk}(\theta) = \begin{bmatrix} \mathrm{U}_{nlk}(\theta) \\ i\,\mathrm{V}_{nlk}(\theta) \\ \Phi_{nlk}(\theta) \end{bmatrix}, \tag{30}$$

where the factor $i$ in front of $\mathrm{V}_{nlk}$ is introduced to account for a phase shift of $\pi/2$ (Kasahara, 1977; Swarztrauber and Kasahara, 1985). Substituting eq. (30) into $\mathbf{H}_{nlk}(\lambda,\theta) = \mathbf{\Theta}_{nlk}(\theta)\,e^{in\lambda}$ in eq. (29), one can see that the Hough vector functions satisfy the orthonormal condition given by

$$\int\limits_{-\frac{\pi}{2}}^{\frac{\pi}{2}} \mathbf{\Theta}^*_{nlk} \cdot \mathbf{\Theta}_{nl'k} \cos\theta \mathrm{d}\theta = \int\limits_{-\frac{\pi}{2}}^{\frac{\pi}{2}} (\mathrm{U}_{nlk}\,\mathrm{U}_{nl'k} + \mathrm{V}_{nlk}\,\mathrm{V}_{nl'k} + \Phi_{nlk}\,\Phi_{nl'k}) \cos\theta \mathrm{d}\theta = \delta_{ll'}. \tag{31}$$

Using orthonormal condition (29), a complete set of Fourier-Hough transforms may be constructed as

$$\mathbf{W}_k(\lambda,\theta,t) = \sum_{l=0}^{\infty} \sum_{n=-\infty}^{\infty} w_{nlk}(t)\,\mathbf{H}_{nlk}(\lambda,\theta), \tag{32}$$

$$w_{nlk}(t) = \frac{1}{2\pi} \int\limits_{-\frac{\pi}{2}}^{\frac{\pi}{2}} \int\limits_{0}^{2\pi} \mathbf{H}^*_{nlk}(\lambda,\theta) \cdot \mathbf{W}_k(\lambda,\theta,t) \cos\theta \mathrm{d}\lambda \mathrm{d}\theta. \tag{33}$$

In order to distinguish each wave type, i.e. the westward–propagating Rossby wave and the westward– and eastward–propagating gravity-inertia waves, the meridional index $l$ is defined as a sequence of three distinct modes, $l_r$, $l_w$ and $l_e$,
respectively (e.g. Kasahara and Puri, 1981; Marques and Castanheira, 2012).





## 2.4 Solutions of the horizontal structure equation

As shown in Swarztrauber and Kasahara (1985), the meridional modal functions $\mathbf{\Theta}_{nlk}(\theta)$ for the eastward and westward gravity-inertia modes are symmetric (antisymmetric) with respect to the equator for even (odd) meridional index $l$. For the rotational modes, the functions $\mathbf{\Theta}_{nlk}(\theta)$ are symmetric (antisymmetric) for odd (even) $l$. For each vertical mode $k$, the fre-
quencies $\sigma_{nlk}$ and the corresponding meridional modal functions $\mathbf{\Theta}_{nlk}(\theta)$ are computed for a range of zonal wavenumbers $n$ and meridional indices $l$. The computational method was developed by Swarztrauber and Kasahara (1985) and consists of expanding the eigensolutions of (28) in terms of spherical vector harmonics. The modes for $n = 0$ are determined following the approach suggested by Shigehisa (1983), as described in Swarztrauber and Kasahara (1985).

Figure 4 shows the dimensionless frequencies $\sigma$ of westward propagating gravity-inertia and rotational waves (left panel)
and eastward propagating gravity-inertia waves (right panel), for the case $n = 1$. This figure corresponds to Figure 2 in Swarztrauber and Kasahara (1985), because we have used the same values of $\alpha$ has they did to compute the frequencies $\sigma$. The $\alpha$ used here corresponds to $1/\sqrt{\epsilon}$ of Swarztrauber and Kasahara (1985). The frequencies are plotted as a function of $\alpha^{-1}$ for 10 meridional indices ($l = 0, \ldots, 9$), five symmetric (continous lines) and five antisymmetric (dashed lines). The logarithm scale was used in both axis and the $y$-axis for the westward propagating waves has been reversed. The rotational mode $l_r = 0$
is referred to as a mixed Rossby-Gravity wave because it behaves like a rotational mode for large values of $\alpha$, but behaves like a gravity mode for small values of $\alpha$. The symmetric mode $l_e = 0$ is referred to as a Kelvin wave.

The meaning of eastward and westward propagations is lost in the case of $n = 0$. However, since the frequencies of gravity-inertia motion appear as pairs of positive and negative values with the same magnitudes, we also use the term eastward (westward) to indicate positive (negative) frequency, as adopted by Swarztrauber and Kasahara (1985). The frequencies of
the rotational motion along with the frequencies of the gravity motion corresponding to the lowest meridional index ($l = 0$) are zero for $n = 0$ (Swarztrauber and Kasahara, 1985). Therefore, instead of saving zeros for all the rotational frequencies and for the gravity frequencies corresponding to the lowest meridional index, the software computes the asymptotic rate ($\sigma_a$) at which those frequencies go to zero (see eqs. (4.14) and (4.18) in Swarztrauber and Kasahara, 1985, for the definition and computational formula of $\sigma_a$).

Left panel of Figure 5 shows the curves of asymptotic dimensionless frequencies $\sigma_a$ in the case of $n \to 0$. The frequencies are plotted as a function of $\alpha^{-1}$ for 10 meridional indices, using a log-log scale with the $y$-axis reversed. In this case, the lowest symmetric mode is identified as $l_r = -1$ because the symmetric equations begin at index $-1$ (see Swarztrauber and Kasahara, 1985, for details). The values of $\sigma_a$ are all negative, except for the mode $l_r = -1$, which is similar to the eastward propagating Kelvin wave in Figure 4 (right panel). For this reason, the $l_r = -1$ mode is saved as the eastward gravity mode $l_e = 0$ as in
Swarztrauber and Kasahara (1985). The curves for the modes $l_r = 1, 2, \ldots, 9$ behave analogously to those of rotational modes in Figure 4 (left panel). The absolute values of dimensionless frequencies $\sigma$ in the case of $n = 0$ of the gravity waves, are illustrated in the right panel of Figure 5 for $l = 1, 2, \ldots, 9$. The right and left panels of Figure 5 corresponds to Figures 1 and 3 of Swarztrauber and Kasahara (1985), respectively.







**Figure 4.** Dimensionless frequencies $\sigma$ for westward propagating gravity waves and rotational waves (left) and for eastward propagating gravity waves (right), plotted as a function of $\alpha^{-1}$ and for zonal wavenumber $n = 1$.



**Figure 5.** Asymptotic dimensionless frequencies $\sigma_a$ in the case of $n \to 0$ (left) and dimensionless frequencies $|\sigma|$ for gravity waves and zonal wavenumber $n = 0$ (right), plotted as a function of $\alpha^{-1}$.





As mentioned above, the horizontal modes include Kelvin, westward and eastward inertio–gravity (WIG and EIG, respectively), Rossby and mixed Rossby–gravity (MRG) waves. As an illustration of the software outputs, the horizontal structure functions were calculated for each equivalent height computed in subsection 2.2, using boundary conditions (14) and (15) (the first 12 VSFs are represented in Figure 2), and for 43 wavenumbers ($n = 0, \dots, 42$) and 80 meridional modes (20 WIG, 20 EIG and 40 Rossby modes). The meridional profiles of the Hough functions corresponding to two Rossby modes $l_r = 1$ and $l_r = 2$ are shown in Fig. 6 for the barotropic mode ($k = 0$, with $h_0 \simeq 9.8$ km) and for two zonal wavenumbers ($n = 0$ and $n = 10$). Figure 7 shows the meridional profiles corresponding to the Kelvin mode ($l_e = 0$) and to the Rossby mode ($l_r = 1$) for two baroclinic modes ($k = 1$, with $h_1 \simeq 6.1$ km and $k = 10$, with $h_{10} \simeq 65$ m) and for two zonal wavenumbers ($n = 0$ and $n = 10$). Increasing the vertical index $k$ (i.e. reducing the equivalent height $h_k$) or increasing the zonal wavenumber $n$ for the same vertical index $k$, the horizontal structure functions become more confined around the equator (Longuet-Higgins, 1968; Cohn and Dee, 1989; Žagar et al., 2015b).

### 2.4.1 The Haurwitz waves

The Hough vector functions define modes in the transformed variables as given by eq. (23). If the equivalent height $h_k$ is infinite, then transform (23) is no longer applicable. This is the case for the barotropic mode when the lower boundary condition (16) is used. However, an infinite equivalent height means that the separation constant in eq. (12) is null, and the horizontal motion is non-divergent, and there must be no vertical dependence in the linear system (8)–(10). Consistently, the respective vertical structure is a constant. In this case, the horizontal modes are computed in the untransformed variables $u, v$ and $\phi$ as in Swarztrauber and Kasahara (1985), and are called the Rossby-Haurwitz waves. Additionally, because linear system (8)–(10) is non divergent there are no WIG and EIG modes (and also no MRG mode for the zonal mean component $n = 0$). Fig. 8 shows the same meridional profiles as in Fig. 6, but for the Haurwitz modes.

### 2.5 3–D Normal mode functions

The three-dimensional (3–D) *normal mode functions* (NMFs) are obtained by the product of the vertical normal modes $\Psi_k(p)$ – the eigensolutions of the VSE – and the horizontal normal modes $\mathbf{H}_{nlk}(\lambda, \theta)$ – the eigensolutions of the HSE (e.g. Kasahara, 1976; Kasahara and Puri, 1981). The 3–D NMFs, denoted by $\Pi_{nlk}(\lambda, \theta, p)$, form a complete orthogonal basis, allowing therefore to expand the horizontal wind and the geopotential fields of the global atmosphere (Kasahara and Puri, 1981; Tanaka, 1985; Daley, 1991):

$$(u, v, \phi)^{\mathrm{T}} = \sum_{k=0}^{\infty} \sum_{l=0}^{\infty} \sum_{n=-\infty}^{\infty} w_{nlk}(t) \, \mathbf{X}_k \, \mathbf{\Pi}_{nlk}(\lambda, \theta, p), \tag{34}$$

where

$$\mathbf{\Pi}_{nlk}(\lambda, \theta, p) = \Psi_k(p) \, \mathbf{\Theta}_{nlk}(\theta) \, e^{i n \lambda}. \tag{35}$$





**Figure 6.** Meridional structures of the barotropic ($k = 0$, $h_0 \simeq 9.8$ km) Hough harmonics for two Rossby modes, $l_r = 1$ (left panels) and $l_r = 2$ (right panels), and two zonal wavenumbers, $n = 0$ (top panels) and $n = 10$ (bottom panels). The equivalent height $h_0$ was obtained by solving the vertical structure equation with boundary conditions (14) and (15).





**Figure 7.** Meridional structures of two baroclinic Hough harmonics ($k = 1$, $h_1 \simeq 6.1$ km and $k = 10$, $h_{10} \simeq 65$ m) for (left panels) the Kelvin mode, $l_e = 0$ and (right panels) the Rossby mode, $l_r = 1$, and also for two zonal wavenumbers, $n = 0$ and $n = 10$. The equivalent heights $h_k$ were obtained by solving the vertical structure equation with boundary conditions (14) and (15).





**Figure 8.** As in Fig 6, but for the Haurwitz modes ($h_0 = \infty$).





The expansion coefficients $w_{nlk}$ are obtained by means of the vertical projection onto the vertical structure functions, followed by the horizontal projection onto the horizontal structure functions:

$$w_{nlk}(t) = \int\limits_{-\frac{\pi}{2}}^{\frac{\pi}{2}} \int\limits_{0}^{2\pi} \int\limits_{0}^{p_s} \mathbf{\Pi}_{nlk}^{*}\left(\lambda,\theta,p\right) \cdot \mathbf{X}_k^{-1} \mathbf{W} \cos\theta \,\mathrm{d}p\,\mathrm{d}\lambda\,\mathrm{d}\theta, \tag{36}$$

with $\mathbf{W} = [u,\,v,\,\phi]^T$.

## 2.6 3–D normal mode energetics scheme

Using the 3–D NMFs of the linearized primitive equations ((8) – (10)) as a basis to expand the global circulation field, Tanaka (1985) (see also Tanaka and Kung, 1988) developed a 3–D normal mode energetics (NME) scheme, which combines three one-dimensional spectral energetics in domains of zonal wavenumber, $n$, meridional mode number, $l$, and vertical mode number, $k$. The 3–D NME scheme complements therefore the standard energetics in the zonal wavenumber domain of Saltzman (1957), since it can diagnose the 3–D spectral distribution of energy and energy interactions, the energetics characteristics of Rossby waves and gravity-inertia waves, and the energy interaction between the barotropic and baroclinic modes (Tanaka and Kung, 1988).

Derivation of the 3–D NME may be summarized as follows:

Applying the vertical transform (18) to equations (1), (2) and (7), a dimensionless equation is obtained in the following vector form:

$$\frac{\partial}{\partial \hat{t}} \mathbf{W}_k + \mathbf{L}\,\mathbf{W}_k = \mathbf{I}_k + \mathbf{J}_k + \mathbf{S}_k, \tag{37}$$

where subscript $k$ denotes the $k^{th}$ component of the vertical transform and $\hat{t}$, $\mathbf{W}_k$ and $\mathbf{L}$ are given by (22), (23) and (24), respectively. The dimensionless nonlinear term vectors for the wind and mass fields, $\mathbf{I}_k$ and $\mathbf{J}_k$, and the energy source or sink term due to diabatic heating and dissipation, $\mathbf{S}_k$, are given by

$$\mathbf{I}_k = \mathbf{Y}_k^{-1} \begin{bmatrix} -\boldsymbol{V}\cdot\nabla u - \omega\frac{\partial u}{\partial p} + \frac{\tan\theta}{a}uv \\ -\boldsymbol{V}\cdot\nabla v - \omega\frac{\partial v}{\partial p} - \frac{\tan\theta}{a}u^2 \\ 0 \end{bmatrix}_k, \tag{38}$$

$$\mathbf{J}_k = \mathbf{Y}_k^{-1} \begin{bmatrix} 0 \\ 0 \\ \frac{\partial}{\partial p}\left[\frac{R}{pS_0}\left(-\boldsymbol{V}\cdot\nabla T - \omega\frac{\partial T}{\partial p}\right)\right] \end{bmatrix}_k, \tag{39}$$

$$\mathbf{S}_k = \mathbf{Y}_k^{-1} \begin{bmatrix} F_u \\ F_v \\ \frac{\partial}{\partial p}\left(\frac{qR}{pc_pS_0}\right) \end{bmatrix}_k, \tag{40}$$





These vectors are made dimensionless using the inverse of scaling matrix $\mathbf{Y}_k$, which is given by

$$\mathbf{Y}_k = \begin{bmatrix} 2\Omega\sqrt{gh_k} & 0 & 0 \\ 0 & 2\Omega\sqrt{gh_k} & 0 \\ 0 & 0 & 2\Omega \end{bmatrix}. \tag{41}$$

Applying the Fourier-Hough transform (33) to (37), yields

$$\frac{\mathrm{d}}{\mathrm{d}t} w_{nlk} + i\,\sigma_{nlk}\,w_{nlk} = \imath_{nlk} + \jmath_{nlk} + s_{nlk}, \tag{42}$$

where the complex variables $w_{nlk}$, $\imath_{nlk}$, $\jmath_{nlk}$ and $s_{nlk}$ are the Fourier-Hough transforms of the vector variables $\mathbf{W}_k$, $\mathbf{I}_k$, $\mathbf{J}_k$ and $\mathbf{S}_k$, respectively.

The kinetic energy, $K$, and available potential energy, $A$, are defined respectively by

$$K = \frac{1}{2}\left(u^2 + v^2\right), \tag{43}$$

$$A = \frac{1}{2\,S_0}\left(\frac{\partial\phi}{\partial p}\right)^2, \tag{44}$$

And the total energy, $E = K + A$, is conserved a quantity of the full nonlinear equations (1), (2) and (7) under frictionless and adiabatic conditions (see Tanaka and Kung, 1988).

In the transformed 3-D normal mode space, the total energy $E_{nlk}$ associated with each mode $(nlk)$ is

$$E_{nlk} = \frac{1}{2\,\epsilon_n} p_s\,h_k|w_{nlk}|^2. \tag{45}$$

where $\epsilon_n = 2$ for the zonal $(n = 0)$ modes, and $\epsilon_n = 1$ for the eddy $(n > 0)$ modes.

Substituting (42) into the time derivatives of (45), the energy balance equations for the 3–D normal modes are finally obtained as

$$\frac{\mathrm{d}}{\mathrm{d}t} E_{nlk} = I_{nlk} + J_{nlk} + S_{nlk}, \tag{46}$$

where

$$I_{nlk} = \frac{p_s\,\Omega\,h_k}{\epsilon_n}\left[w_{nlk}^*\,\imath_{nlk} + w_{nlk}\,\imath_{nlk}^*\right], \tag{47}$$

$$J_{nlk} = \frac{p_s\,\Omega\,h_k}{\epsilon_n}\left[w_{nlk}^*\,\jmath_{nlk} + w_{nlk}\,\jmath_{nlk}^*\right], \tag{48}$$

$$S_{nlk} = \frac{p_s\,\Omega\,h_k}{\epsilon_n}\left[w_{nlk}^*\,s_{nlk} + w_{nlk}\,s_{nlk}^*\right]. \tag{49}$$

The terms $I_{nlk}, J_{nlk}$, and $S_{nlk}$ that contribute to the time change of $E_{nlk}$ are respectively associated with nonlinear interactions of kinetic and available potential energies and with an energy source or sink due to diabatic heating and dissipation. The spetra of these terms can therefore be analysed separately for the zonal mean and eddy components, for the barotropic and baroclinic modes, and also for the Rossby and gravity waves (Tanaka and Kung, 1988).





As another illustration of the capabilities of the current software, in Figure 9 it is shown the spectra for total energy $E$ (top panels), interactions of available potential energy $J$ (middle panels) and interactions of kinetic energy $I$ (bottom panels). The left column of panels contain the wavenumber spectra whereas the middle and right columns show the vertical spectra for the zonal mean and eddy components, respectively. In each panel it is represented the total spectra along with the contributions

of the Rossby and Gravity components. These spectra were computed using 34 years of ERA-interim reanalysis (Dee et al., 2011) data covering the period from 1 January, 1979 to 31 December, 2012. The horizontal wind $(u, v)$, pressure velocity $(\omega)$, temperature and geopotential fields were obtained with $1.5°$lon. $\times 1.5°$lat. grid resolution for the 37 isobaric levels from 1000- to 1 hPa, at time intervals of six-hours. The solutions of the VSE ((13)), with boundary conditions (16) and (15), were approximated using 57 Legendre polynomials and 37 vertical modes were retained. The zonal wavenumber has been truncated

at $n = 42$. The spectra were computed at each time step (6-hourly) and then averaged over the 30 years period.

A detailed analysis of the spectra in figure 9 is beyond the scope of this study, but some of its characteristics are mentioned. The wavenumber energy spectrum of Rossby modes follows approximately the -3 power law over the synoptic and mesoscale region, while that of the gravity modes has approximately a -5/3 slope, which is in line with the literature (e.g. Charney, 1971; Nastrom and Gage, 1985; Tanaka, 1985; Tanaka et al., 1986; Terasaki and Tanaka, 2007). The zonal mean available potential

energy is transferred into eddy available potential energy essentially due to the Rossby waves, with a small contibution by planetary scale gravity waves, as indicated by the positive values in the wavenumber spectra of $J$. On the other hand, the wavenumber spectra of $I$ indicate that the Rossby waves transfer energy out of the eddy kinetic energy reservoir, whereas the smaller quantity of energy transferred by the gravity waves is in the opposite direction. The vertical spectra for $I$ shows that the eddy kinetic energy contained in the Rossby baroclinic modes is transferred into the Rossby barotropic modes of both zonal

mean and eddy components, and also into baroclinic zonal mean kinetic energy. Again, it is seen that the transfer of kinetic energy due to the gravity waves is in the opposite direction.

A disadvantage of the 3–D NME is that one cannot separate the available potential and kinetic energies, only the total energy $(E_{nlk})$ associated with each mode can be calculated. Consequently, also the conversion rate of available potential energy into kinetic energy cannot be acessed. However, the separation of the available potential and kinetic energies can be performed if

one uses only the expansions in the vertical and wavenumber domains. This reasoning led Marques and Castanheira (2012) to present a normal mode energetics formulation that performs an explicit evaluation of the available potential and kinetic energies as well as the conversion rates between them. In addition, the generation and dissipation rates and also the nonlinear interactions of each energy form, can be performed in both the zonal wavenumber and vertical mode domains. Using the vertical normal mode basis that results from applying lower boundary condition (14) and (15), and then performing the vertical

and the zonal wavenumber (Fourier) expansions, the set of balance equations for the kinetic energy $(K)$ and available potential energy $(A)$ is given by (see Marques and Castanheira, 2012, for details)

$$\frac{\mathrm{d}K_{nk}}{\mathrm{d}t} = C_{nk} + I_{nk} - D_{nk} \tag{50}$$

$$\frac{\mathrm{d}A_{nk}}{\mathrm{d}t} = -C_{nk} + J_{nk} + G_{nk}, \tag{51}$$



**Figure 9.** Spectra of total energy $E$ (top panels) and of nonlinear interactions of available potential $J$ (middle panels) and kinetic $I$ (bottom panels) energies, in both the wavenumber (left column) and the vertical (middle column for the zonal component (0) and right column for the eddies (n)) domains. In each panel it is represented the contributions of the Rossby (R) and Gravity (G) components for the total (T) spectra of energy or energy interactions.





where

$$K_{nk} = \frac{p_s\,h_k}{2\,\epsilon_n} \int\limits_{-\frac{\pi}{2}}^{\frac{\pi}{2}} \left( \hat{u}_{nk}^*\,\hat{u}_{nk} + \hat{v}_{nk}^*\,\hat{v}_{nk} \right) \cos\theta \mathrm{d}\theta, \tag{52}$$

$$A_{nk} = \frac{p_s\,h_k}{2\,\epsilon_n} \int\limits_{-\frac{\pi}{2}}^{\frac{\pi}{2}} \left( \hat{\phi}_{nk}^*\,\hat{\phi}_{nk} \right) \cos\theta\,\mathrm{d}\theta, \tag{53}$$

$$C_{nk} = \frac{p_s\,\Omega\,h_k}{\epsilon_n} \int\limits_{-\frac{\pi}{2}}^{\frac{\pi}{2}} \left\{ \frac{i\,n\,\alpha_k}{\cos\theta} \left( \hat{u}_{nk}\,\hat{\phi}_{nk}^* - \hat{u}_{nk}^*\,\hat{\phi}_{nk} \right) \right.$$
$$\left. - \alpha_k \left( \hat{v}_{nk}^*\,\frac{\partial\hat{\phi}_{nk}}{\partial\theta} + \hat{v}_{nk}\,\frac{\partial\hat{\phi}_{nk}^*}{\partial\theta} \right) \right\} \cos\theta\,\mathrm{d}\theta, \tag{54}$$

$$I_{nk} = \frac{p_s\,\Omega\,h_k}{\epsilon_n} \int\limits_{-\frac{\pi}{2}}^{\frac{\pi}{2}} \left\{ \left( \hat{u}_{nk}^*\left(\hat{I}_1\right)_{nk} + \hat{u}_{nk}\left(\hat{I}_1\right)_{nk}^* \right) \right.$$
$$\left. + \left( \hat{v}_{nk}^*\left(\hat{I}_2\right)_{nk} + \hat{v}_{nk}\left(\hat{I}_2\right)_{nk}^* \right) \right\} \cos\theta\,\mathrm{d}\theta, \tag{55}$$

$$J_{nk} = \frac{p_s\,\Omega\,h_k}{\epsilon_n} \int\limits_{-\frac{\pi}{2}}^{\frac{\pi}{2}} \left( \hat{\phi}_{nk}^*\left(\hat{J}_3\right)_{nk} + \hat{\phi}_{nk}\left(\hat{J}_3\right)_{nk}^* \right) \cos\theta\,\mathrm{d}\theta, \tag{56}$$

$$D_{nk} = -\frac{p_s\,\Omega\,h_k}{\epsilon_n} \int\limits_{-\frac{\pi}{2}}^{\frac{\pi}{2}} \left\{ \left( \hat{u}_{nk}^*\left(\hat{F}_u\right)_{nk} + \hat{u}_{nk}\left(\hat{F}_u\right)_{nk}^* \right) \right.$$
$$\left. + \left( \hat{v}_{nk}^*\left(\hat{F}_v\right)_{nk} + \hat{v}_{nk}\left(\hat{F}_v\right)_{nk}^* \right) \right\} \cos\theta\,\mathrm{d}\theta, \tag{57}$$

$$G_{nk} = \frac{p_s\,\Omega\,h_k}{\epsilon_n} \int\limits_{-\frac{\pi}{2}}^{\frac{\pi}{2}} \left( \hat{\phi}_{nk}^*\left(\hat{N}_3\right)_{nk} + \hat{\phi}_{nk}\left(\hat{N}_3\right)_{nk}^* \right) \cos\theta\,\mathrm{d}\theta, \tag{58}$$

15 and

$$\left(\hat{I}_1\right)_{nk} = \frac{1}{2\Omega\sqrt{g\,h_k}} \left[ -\boldsymbol{V}\cdot\nabla u - \omega\frac{\partial u}{\partial p} + \frac{\tan\theta}{a}uv \right]_{nk}, \tag{59}$$

$$\left(\hat{I}_2\right)_{nk} = \frac{1}{2\Omega\sqrt{g\,h_k}} \left[ -\boldsymbol{V}\cdot\nabla v - \omega\frac{\partial v}{\partial p} - \frac{\tan\theta}{a}u^2 \right]_{nk}, \tag{60}$$

$$(\hat{J}_3)_{nk} = \frac{1}{2\Omega} \left( \frac{\partial}{\partial p}\left[ \frac{R}{pS_0}\left( -\boldsymbol{V}\cdot\nabla T - \omega\frac{\partial T}{\partial p} \right) \right] \right)_{nk}, \tag{61}$$

$$(\hat{N}_3)_{nk} = \frac{1}{2\Omega} \left[ \frac{\partial}{\partial p}\left( \frac{qR}{p\,c_p\,S_0} \right) \right]_{nk}, \tag{62}$$


$$\left(\hat{F}_u\right)_{nk} = \frac{(F_u)_{nk}}{2\,\Omega\,\sqrt{g\,h_k}},$$ (63)

$$\left(\hat{F}_v\right)_{nk} = \frac{(F_v)_{nk}}{2\,\Omega\,\sqrt{g\,h_k}}.$$ (64)

Term $C_{nk}$ represents the conversion of available potential energy into kinetic energy, whereas $I_{nk}$ ($J_{nk}$) represents interac-

tions or exchanges of kinetic (available potential) energy between different scales or types of motion. Finally, the dissipation

of kinetic energy and the generation of available potential energy, due to diabatic processes, are represented by terms $D_{nk}$ and

$G_{nk}$, respectively, and may be computed as residuals from balance equations (50) and (51). The calculated energetics terms

may then be used to construct an extended energy cycle diagram, as in Fig. 7 of Marques and Castanheira (2012) (see also

Fig. 6 in Marques et al., 2014). Such an extended energy cycle diagram is illustrated in figure 10, in which the boxes represent

the levels of energy and the arrows the energy generation/dissipation rates and the energy conversion/transfer rates. The esti-

mates in the diagram were based on the same 34 years of ERA-interim reanalysis mentioned above for the 3-D spetra of $E$,

$J$ and $I$. The solutions of the VSE ((13)) were obtained using boundary conditions (14) and (15). The energetics terms were

computed at each time step (6-hourly) and then averaged over the 34 years period to obtain the corresponding mean values

for the northern winter (DJF) and summer (JJA) seasons, which are represented as the top (black) and bottom (red) values in

figure 10.

The energy cycle diagram of figure 10 describes the flow of energy among the zonal mean and eddy components, and

also among the barotropic and baroclinic components. All terms in the diagram are decomposed into the zonal mean ($n =$

0) and eddy ($n \geq 1$) components, which are denoted respectively by subscripts $Z$ and $E$. Each one of these components

is also decomposed into the barotropic ($k = 0$) and baroclinic ($k \geq 1$) components, by using the extra subscripts $B$ and $b$,

respectively. The two exceptions are the barotropic-baroclinic interactions of available potential energy, denoted by $J_{Bb}$, and

the baroclinic-barotropic interactions of kinetic energy which is represented as $I_{bB}$. These values represent the balances of the

flows of kinetic energy and available potential energy between the barotropic and baroclinic components and both zonal mean

and eddies, implying that the energy flows represented by the dotted lines in the diagram cannot by quantified individually.

These flows are associated with eddy generation by barotropic instability and the barotropic decay of baroclinic eddies (see

Marques and Castanheira, 2012, and references therein, for details)

As an alternative, Marques and Castanheira (2017) and Castanheira and Marques (2019) presented a similar NME scheme

but using the vertical normal mode basis that results from applying lower boundary condition (16) in place of (14). The main

difference between the NME schemes presented in Marques and Castanheira (2012) and in Marques and Castanheira (2017)

is that in the later, the first vertical structure is strictly constant, which implies that its vertical derivative is null and there is no

available potential energy associated with the barotropic mode ($k = 0$). Therefore, the terms $A_{n0}$, $C_{n0}$, $J_{n0}$ and $G_{n0}$ are null in

this variant of the energetics scheme. All baroclinic terms ($k \geq 1$) have identical expressions in both NME schemes. The terms

associated with the kinetic energy of the barotropic component are calculated formally with identical expressions in both NME

schemes. However, a constant $h_0$ can not be derived from de eigenvalues of the VSE (13). In this case the smallest eigenvalue

is null, and corresponds to the separation constant in eq. (12) to be null. For such case, it is not possible to define an equivalent





**Figure 10.** Extended energy cycle diagram describing the flow of energy among the zonal mean and eddy components, and also among the barotropic and baroclinic components for ERA-Interim in DJF (top values) and JJA (bottom values) climates. Units are $10^5$ J m$^{-2}$ for energy levels and W m$^{-2}$ for conversion/transfer rates and generation/dissipation rates.



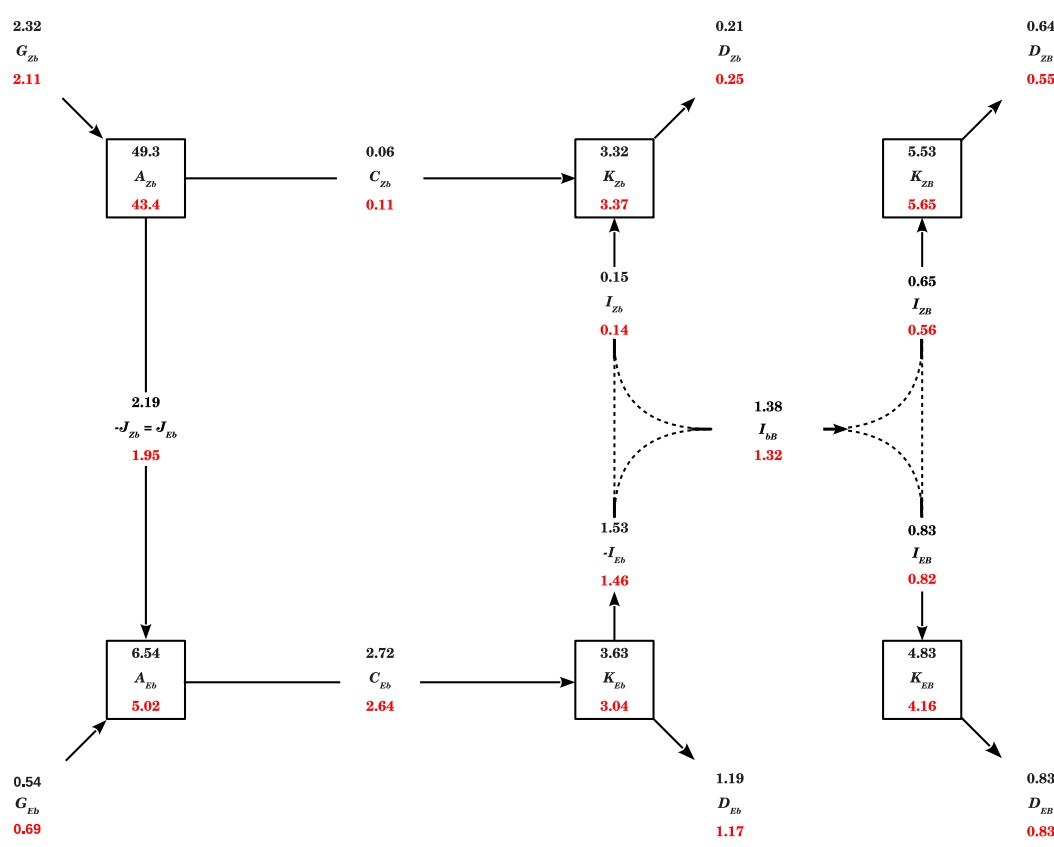

**Figure 11.** As in Fig. 10, but replacing lower boundary condition (14) by (16).

height as in eq. (12). In order to maintain the expressions for the barotropic and baroclinic terms formally identical in both NME schemes, the software fixes $h_0 = 1m$, when the boundary condition (16) is used. Performing in this way the equations (50) and (51), with its individual terms given by equations (52)-(58), may be also used to construct an extended energy cycle diagram as illustrated in figure 11 (see also Fig. A2 in Marques and Castanheira, 2017). In this case, there is no barotropic

5    branch in the available potential energy side, as compared to extended energy cycle diagram of figure 10. The estimates in this diagram were based on the same data as those of figure 10. The solutions of the VSE were also approximated using 57 Legendre polynomials, being retained 37 vertical modes. It was also considered the same zonal wavenumbers as those for the estimates in figure 10.





Although boundary condition (16) may be seen as somewhat unrealistic, it presents some useful features. First, using the boundary conditions (15) and (16), it can be shown easily that the total energy is exactly conserved in by full non-linear equations (1)-(5), in the case of frictionless adiabatic motions (see Tanaka and Kung, 1988). Moreover, an extra surface term, which was interpreted by Tanaka and Kung (1988) as a geopotential flux across the lower boundary, does not appear in the available potential energy as defined by eq. (44), when the boundary condition (16) is used. Another feature of (16) is that the barotropic mode represents the mass weighted average of the circulation field.

The use of the boundary condition (14) may seem more realistic, but in order to guaranty the conservation of energy for frictionless adiabatic motions it is necessary to impose a non-slipping condition $(u, v)_s = 0$ at $p = p_s$. And this may seem contradictory with the absence of friction.

## 3   Conclusions

The software here presented consists of tools or functions for specific tasks and includes a Python and a Matlab version. The Python version requires Numpy and Scipy modules and works with both Python 2 and 3. The netCDF4 library is recommended. The input/output data format for each function may be either the native format of the chosen version, i.e. ".mat" for Matlab and ".npz" for Python, or netCDF for both versions.

The tasks to be achieved with the functions in the software include the solution of the VSE (13), the solution of HSE (28) and the computation of complex expansion coefficients $w_{nlk}$, $\imath_{nlk}$ and $\jmath_{nlk}$, which are the vertical-Fourier-Hough transforms of the dependent variable vector $[u, v, \phi]^T$, of the nonlinear term vector due to wind field and of the nonlinear term due to mass field, respectively. In addition, the software also permits the computation of vertical transforms of the terms in the normal mode energetics formulation presented by Marques and Castanheira (2012) or its alternative presented by Marques and Castanheira (2017) and Castanheira and Marques (2019). These terms constitute the set of balance equations (50) and (51) and its expressions are given in equations (52)-(64). With these vertical transforms the user may compute the global energy cycle in the wavenumber and vertical domains (as in Marques and Castanheira, 2012, 2017; Castanheira and Marques, 2019), with the dissipation and generation terms computed as residuals from (50) and (51).

Other applications of this software include the spatial scale filtering atmospheric motion, and filtering of balanced motion and mass unbalanced motions by retaining an appropriate subset of terms in the expansion (34) (e.g. Castanheira and Marques, 2015; Žagar et al., 2019). Furthermore, the solutions of HSE, i.e. the Hough vectors functions $(\Theta_{nlk}(\theta))$, can be used as meridional basis functions onto which dynamical data may be projected. This can be an alternative tool for investigating studies on tropical convection as in Yang et al. (2003) and Gehne and Kleeman (2012), which have used parabolic cylinder functions as meridional basis functions. All these features are useful both in data analysis and in assessment of general circulation atmospheric models (e.g. Žagar et al., 2019).





**Appendix A:  Software functions**

- **Vertical structure**

  The vertical structure equation (VSE), (13), is solved with function named *vertical_structure* that uses the spectral method introduced by Kasahara (1984). The function requires the mean vertical profile of temperature $T_0$ , defined on pressure levels, and the pressure levels as inputs, returning the vertical structure functions ($\Psi_k(p)$) and equivalent heights ($h_k$). The VSE may be solved using either one of the two pairs of boundary conditions as described in section 2.2. These pairs of boundary conditions are controled by the input argument *ws0*. When *ws0=False* (the default) the VSE is solved using boundary conditions (14) and (15), whereas when *ws0=True* the VSE is solved using boundary conditions (16) and (15). In the later case the first vertical structure is strictly constant and the associated equivalent height is infinite.

- **Horizontal structure**

  The solution of HSE (21) is obtained with function called *hough_functions*. It uses the computational method developed by Swarztrauber and Kasahara (1985) which consists of expanding the eigensolutions of (28) in terms of spherical vector harmonics. The function inputs are the equivalent heights ($h_k$) and the user defined parameters for the maximum zonal wavenumber, the total number of Rossby modes and half the number of Gravity modes used in the expansion, along with the latitude points which may be linear (the default) or gaussian. It returns the Hough vectors functions ($\Theta_{nlk}(\theta)$) and the frequencies of the modes. In the output, the meridional modes are written in the following order: first the westward gravity modes, then the eastward gravity modes and finally the (westward) Rossby modes. Both types of gravity modes are written following the symmetric-antisymmetric mode order, whereas the order for Rossby modes is reversed. If the VSE (13) is solved using boundary conditions (16) and (15), then the solutions for the barotropic mode returned by *hough_functions* are the Haurwitz waves (Swarztrauber and Kasahara, 1985). This is controled internaly by *hough_functions* which calls functions named *hvf_baroclinic* and *hvf_barotropic* for the Haurwitz modes.

- **Expansion coefficients**

  The complex expansion coefficients $w_{nlk}$, $\imath_{nlk}$ and $\jmath_{nlk}$ may be obtained with function called *expansion_coeffs*. The inputs are the vertical structure functions ($\Psi_k(p)$), the equivalent heights ($h_k$), the Hough vectors functions ($\Theta_{nlk}(\theta)$) and a data structure with appropriate fields. For the expansion coefficients $w_{nlk}$ it is required a data structure with fields $[u,v,\phi]$ (recall that field $\phi$ corresponds to the deviations from a hydrostatic reference state $\phi_0(p)$). The computation of expansion coefficients $\imath_{nlk}$ and $\jmath_{nlk}$ requires data structures with fields $[I1,I2]$ and $[J3]$, respectively. Each one of these fields need to be pre-computed by the user and correspond to the terms between square brackets of (59), (60) and (61), respectively (Note that in (61) the vertical derivative is outside the square brackets, and this is accounted in the *expansion_coeffs* function). The returned coefficients, $w_{nlk}$, $\imath_{nlk}$ and $\jmath_{nlk}$, may then be used to compute the 3-D spectrum of total energy ($E_{nlk}$) and of the nonlinear interactions of kinetic ($I_{nlk}$) and available potential ($J_{nlk}$) energies (see Fig. 9), using equations (45), (47) and (48), respectively.

- **The inverse transforms**





The zonal and meridional wind, $u$ and $v$, and geopotential perturbation $\phi$ (from the reference geopotential), as given by equation (34), can be obtained with function *inv_expansion_coeffs*. By choosing an appropriate subset of modes, this function can be used for spatial scale filtering of atmospheric motion, and filtering of balanced and mass unbalanced motions, as well as of barotropic and baroclinic components. Moreover, choosing the appropriate meridional indices the

filtering can be used to isolate tropical components of the atmospheric circulation. The function requires as inputs the vertical structure functions ($\Psi_k(p)$), the equivalent heights ($h_k$), the Hough vectors functions ($\boldsymbol{\Theta}_{nlk}(\theta)$), the complex expansion coefficients ($w_{nlk}$), the pressure levels (in hPa), the longitudes (in degrees) and the wavenumber, meridional and vertical indices. In addition, the user can choose to compute all three fields, $u, v$ and $\phi$ (the default), or only a subset of those, which is controled by the input argument *uvz*.

– **Vertical and Hough transforms**

All functions described above have the same name in both versions, Matlab and Python, difering only in the file extension ("m" and ".py", respectively). The software tools presented here include two more functions used to compute the vertical and the Hough transforms. In the Matlab version there are two separate functions, *vertical_transform.m* and *hough_transform.m*, whereas for Python both are included in module *transforms.py*, being invoked as *transforms.vertical*

and *transforms.hough*. The vertical and Hough transforms are used by the *expansion_coeffs* function, but they may be also executed independently by the user. For example, the *vertical_transform* function (or *transforms.vertical*) may be used to compute the vertical transforms of $u$, $v$, $\phi$, $I1$, $I2$ and $J3$. Again, $I1$, $I2$ and $J3$ correspond to the terms between square brackets of (59), (60) and (61). The computation of these six vertical transforms constitute the essential task to obtain the global energy cycle in the wavenumber and vertical domains as in Marques and Castanheira (2012)

or Marques and Castanheira (2017) (see also Castanheira and Marques (2019)). Having these vertical transforms, all the terms of the normal mode energetics formulation represented by balance equations (50) and (51), may then be easily obtained using equations (52)-(56), with the remainder dissipation and generation terms computed as residuals from (50) and (51).

*Code and data availability.* The exact version of the software code used to produce the results presented in this paper is archived on Zenodo

(Marques et al., 2020). A step by step tutorial is included in the repository for both Python and MATLAB. Also included are the data sets with the mean vertical profiles of temperature and geopotential, computed from 32 years (1979-2010) of the ERA-Interim reanalysis data as described in the text (see section 2.2). These two vertical profiles along with the horizontal wind ($u$, $v$), pressure velocity ($\omega$), temperature and geopotential fields that may be obtained from the ERA-Interim data server with $1.5°$lon. $\times 1.5°$lat. grid resolution for all the provided 37 isobaric levels from 1000- to 1 hPa, at time intervals of six-hours, are the only data sets needed to obtain all the figures in the paper. The

two exceptions are figures 4 and 5 for which we have used the same values as Swarztrauber and Kasahara (1985) (see for example their Table 5. Note that our $\alpha$ corresponds to their $1/\sqrt{\epsilon}$).



*Author contributions.* CAFM conceived the idea, wrote the initial version of the software code for MATLAB and tested the final version of the software for both MATLAB and Python. MMA wrote the software code for Python, the final version of the code for MATLAB and collaborated in the tests of the software. JMC elaborated the structure of the software, collaborated in writing the initial version of the software code for MATLAB and in the software tests. All authors contributed to writing the manuscript.

5 *Competing interests.* The authors declare that no competing interests are present.

*Disclaimer.* The code is made publicly available without any warranty.

*Acknowledgements.* C. A. F. Marques was supported by the Portuguese Foundation for Science and Technology (FCT) under grant SFRH/BPD/76232/201
The ERA-Interim data have been obtained from the ECMWF data server.





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
