# Peer review of "Three-dimensional normal mode functions: Open access tools for their computation in isobaric coordinates (p-3DNMF.v1)"

_Geoscientific Model Development, 2019_

## Referee Comment (RC1) · Anonymous Referee #1 · 16 Mar 2020

The authors present new python and Matlab software that use the normal mode function mathematical formulation to decompose three-dimensional atmospheric fields in vertical pressure coordinates into easterly and westerly propagating inertial gravity waves, and Rossby waves, for each zonal, meridional and vertical scale.

This in itself is not new and there are other codes (e.g. MODES) that do similar things in sigma coordinates. However, from this reviewer's perspective, the new and exciting contribution here is the ability to decompose the nonlinear transfers between scales directly - as opposed to inferring them from the coherence amplitude between the expansion coefficients as done in one of the cited studies.

I found the manuscript well written, that they cited all of the appropriate literature, clearly articulated the mathematical formulation, and presented an adequate cross section of the types of results one can generate using their software. The software is also clearly available for download for other researchers to reproduce their results.

In short I strongly recommend this paper for publication in its current form.
* * *

---

## Referee Comment (RC2) · Anonymous Referee #2 · 8 Apr 2020

In the present paper the authors describe a free software package (p-3DNMV.v1) for computing 3-Dimensional Mormal Modes. The software is based on the original work of Kasahara and Puri (1981), and Tanaka (1988), and subsequent work by different authors. The described software seems comparable to MODES (Žagar et al., 2015) in large parts except the authors use Python and MATLAB instead of FORTRAN, and the computation is done on pressure instead of sigma levels. While the former may make the package more user friendly, the latter allows for computing an energy cycle including the exchanges between kinetic and potential energy.

Overall the paper is well written and structured. It provides a comprehensive descrip-

tion of the theory (equations) including relevant references, and convincingly illustrates the potential of the provided software. In addition, the code is easily accessible, together with a tutorial and some test data (though the data are not part of the software archive, as indicated in 'code and data availability' (p27L25)). Although the paper does not present significant new scientific results nor a new methodology the development and provision of such a software package is a very valuable contribution worth to be published in GMD. Thus, I recommend publication. I only have two remarks, where (2) is of pure technical nature:

1) Reading the abstract, the reader may assume that a new methodology is presented as the basis of the software package. However, as described in the introduction the packaged is mainly based on the original work of Kasahara and Puri (1981), and of Tanaka (1985). This may be made clearer already in the abstract. In addition, in the software manual (tutorial) no reference is made to the work of Tanaka. This, in my view, needs to be changed.

2) Technical: The authors may think of improving Figures 10 & 11 by enlarging the boxes/numbers/characters

---

## Author Comment (AC1) · 5 May 2020

Dear Sir/Madam,

Thank you very much for your positive comments.

---

## Author Comment (AC2) · 5 May 2020

Dear Sir/Madam,

Thank you very much for your comments on our paper. Please see below how we have addressed your comments.

Reviewer's general comment:   In addition, the code is easily accessible, together with a tutorial and some test data (though the data are not part of the software archive, as indicated in 'code and data availability' (p27L25)).

[Figure]

**Response:** In the *'Code and data availibility'* we intended to say that the "mean vertical profiles of temperature and geopotential, computed from 32 years (1979-2010) of the ERA-Interim reanalysis data" were made available. The sentence pointed by the reviewer is followed by the following one: "These two vertical profiles along with the horizontal wind $(u, v)$, pressure velocity $(\omega)$, temperature and geopotential fields that may be obtained from the ERA-Interim data server..."

In order to avoid misunderstandings we removed the words "data sets" from the sentence in p27L25. The sentence is now written as:
"Also included are the mean vertical profiles of temperature and geopotential, computed from 32 years (1979-2010) of the ERA-Interim reanalysis data as described in the text (see section 2.2)."

Reviewer comment # 1: Reading the abstract, the reader may assume that a new methodology is presented as the basis of the software package. However, as described in the introduction the packaged is mainly based on the original work of Kasahara and Puri (1981), and of Tanaka (1985). This may be made clearer already in the abstract. In addition, in the software manual (tutorial) no reference is made to the work of Tanaka. This, in my view, needs to be changed.

**Response:** The abstract does not suggest in any sentence that the package is based in a methodology originally developed by the authors. A misunderstanding as the one pointed by the reviewer is only possible for a reader that has no previous knowledge about the methodology. In such case, the introduction makes clear the previous contributions as it is recognized by the reviewer.

We agree that the software manual (tutorial) should cite the work of Tanaka. The following two references, already included in the manuscript, have been included in the

tutorial:

Tanaka, H. L.: Global energetics analysis by expansion into three-dimensional normal-mode functions during the FGGE winter, J. Meteor. Soc. Japan, 63, 180–200, 1985.

Tanaka, H. L. and Kung, E. C.: Normal-mode energetics of the general circulation during the FGGE year, J. Atmos. Sci., 45, 3723–3736, 1988.

Reviewer comment # 2:    Technical: The authors may think of improving Figures 10 & 11 by enlarging the boxes/numbers/characters.

**Response:** We have followed the suggestion of the reviewer.